# Enable, Reconnect and Augment: A New ERA of Virtual Nature Research and Application

**DOI:** 10.3390/ijerph17051738

**Published:** 2020-03-06

**Authors:** Sigbjørn Litleskare, Tadhg E. MacIntyre, Giovanna Calogiuri

**Affiliations:** 1Faculty of social and health sciences, Inland Norway University of Applied Sciences, 2406 Elverum, Norway; Giovanna.calogiuri@inn.no; 2GO GREEN Initiative, Health Research Institute, University of Limerick, V94 T9PX Limerick, Ireland; Tadhg.macintrye@ul.ie

**Keywords:** virtual reality, technological nature, immersive virtual environments, nature, green exercise, nature based interventions, immersive virtual nature

## Abstract

Being exposed to natural environments is associated with improved health and well-being, as these environments are believed to promote feelings of “being away” from everyday struggles, positive emotional reactions and stress reduction. Despite these positive effects, humanity is becoming increasingly more distanced from nature due to societal changes, such as increased urbanization and the reduced accessibility of natural environments. Technology is also partly to blame, as research suggests that people replace nature contact with increased screen time. In this cross-section between nature and technology, we find technological nature which is progressing towards a point where we may be capable of simulating exposure to real nature. Concerns have been raised regarding this technology, as it is feared it will replace real nature. However, research suggests that virtual nature may have a more positive impact on society than a mere replacement of real nature, and this review propose several areas where virtual nature may be a beneficial addition to actual nature (Enable), help people reconnect with the real natural world (Reconnect) and “boost” human-nature interactions (Augment). Based on the current research and theoretical framework, this review proposes guidelines for future research within these areas, with the aim of advancing the field by producing high quality research.

## 1. Introduction

Human health is influenced by a wide range of factors, including the surrounding environment. Some environments may have a detrimental effect on human health, while others, such as natural environments, are believed to have a salutogenic effect. Research suggests that being exposed to natural environments (e.g., forests, parks and beaches) is associated with improved health and well-being [1,2,3]. For example, White et al. [3] reported positive associations between recreational nature contact in the last seven days and self-reported health and well-being in a sample consisting of almost 20,000 participants. Compared to no nature contact in the previous week, the likelihood of reporting good health or high levels of well-being became significantly greater when participants reported a total duration of nature contact of 120 min per week or more. The health impact of 120 min of nature contact was comparable to achieving the recommended levels of physical activity, living in a high versus low deprivation area or being employed in a high versus low social grade occupation, which signifies the importance of nature contact for public health [3].

Despite the positive effects associated with nature interactions, accumulating evidence suggests that opportunities for nature experiences are decreasing globally [4]. According to the United Nations (UN), more than 55% of the world’s population is currently living in urban environments, and the number is expected to increase to 68% by 2050 [5]. Furthermore, there are reports of a rapid loss of biodiversity and a degradation of the natural world [6] largely due to deforestation and unsustainable land use. In addition to reduced opportunities for nature contact, people are becoming less and less connected with the natural world, a trend that may influence a range of factors such as happiness, life satisfaction, as well as pro-environmental attitudes and behaviors [7,8]. All of this influences how humans perceive and interact with the natural world. In the UK, less than 40% of people visit natural environments during a regular week, of which only a small fraction consists of “active” visits [9]. Even in rural Norway, only 60% of adults engage in active visits in natural environment during a regular week [10]. These numbers are expected to decline even further in the coming years, as today’s children spend less time outside compared to the previous generation [11,12]. Technology is believed to be part of this problem, as research suggests that many replace experiences in nature with increased screen time [12,13]. However, technology might also be part of the solution.

Human-nature interactions are usually associated with the outdoors, but modern technology has enabled people to bring nature experiences into their homes as well. This phenomenon is part of a concept that Peter Kahn defines as technological nature, i.e., technologies that in various ways mediate, augment and simulate our experience of the natural world [14]. The concept of technological nature has received increased attention in the research community. In particular, attention has been directed, often with some concerns, towards so-called virtual nature [15] and especially immersive virtual environments technology [16]. The combination of these two concepts has been defined as immersive virtual nature (IVN) [17], which combines visual and auditory stimuli to create an immersive nature experience. In 1999, when the commercial availability of IVN technology was seen as imminent, Levi and Kocher [15] investigated the potential impact of this technology on society. They found that while IVN may have the advantage of bringing nature to people, as well as increasing people’s support for national natural reserves, this may come at the cost of a devaluation of local natural environments. Others have an even more cautious outlook on our future interactions with nature and fear that technological nature will be a replacement, and a downgrade, of authentic nature in a future where nature contact is severely limited [14]. Completely replacing authentic nature with virtual nature would have severe consequences, as virtual nature lacks the ability to provide important ecosystem services such as climate regulation and nutrient cycling. In spite of these gloomy outlooks on the future of the human species and the potential for interactions with nature, the present paper suggests that technological nature may be more than a mere replacement of real nature and argues that it may rather be part of the solution to increase human-nature interactions and improve public health. In particular, immersive virtual nature may (1) enable us to prolong the positive effects of nature-interactions when we have left the outdoor natural setting; (2) provide access to nature for individuals who may not be able to access it directly; (3) increase feelings of connectedness with the natural world and (4) elicit greater awareness of environmental issues and sustainability. The rapid technological development along with its increased accessibility (and economical affordability) might thus provide a series of opportunities to enhance human-nature interactions.

The purpose of this paper is to provide an overview of the possibilities and challenges of IVN, and offer guidelines for further research. More specifically, this paper will: (I) present an overview of Virtual Reality (VR) technology and IVN and present a general overview of issues associated with IVN technology, (II) discuss three major areas of possible applications or “levels” in which IVN can mediate humans’ interactions with real nature: Enable, Reconnect and Augment (ERA), and (III) propose guidelines for further research based on existing evidence.

## 2. Possibilities and Challenges of Virtual Nature

### 2.1. Virtual Reality and Virtual Nature: Some Definitions

VR can, in short, be referred to as a computer-generated simulation of a three-dimensional image or environment that allows a certain degree of interaction, creating the illusion of reality (See Table 1 for a list of key terms). An important characteristic of VR is that it disconnects the viewer from the external (real) world, which allows the viewer to get immersed in the virtual world. Different types of VR exist, but the one that has gained the most interest in recent years is the type that is enabled by so-called head-mounted displays (HMD), commonly known as “VR masks” or “VR googles”. The introduction of affordable HMDs is a major part of the recent mass appeal of VR technology. HMDs have the advantage of allowing 360° vision of the virtual world while eliminating the visual contact with the external environment. These devices are the basis of the so-called immersive virtual environment technology, which consists of a flow of synthetic sensory information that, through an HMD, provides a surrounding and continuous stream of stimuli, creating the illusory perception of being enclosed within and interacting with a real environment [16,18]. This type of technology is considered more immersive compared to other forms of virtual nature, such as digital images and videos, and is consequently likely to increase the viewers’ sense of presence in the environment. Both immersion and presence are considered key elements of immersive virtual environments as outlined below.

Immersion is solely related to technical aspects of the virtual environment, such as the frame rate, field of view and resolution of the display [25], and one can theoretically evaluate a system’s level of immersion objectively. Presence, on the other hand, describes a person’s subjective feeling of “being in the virtual environment” [24]. This concept relates to the psychological feeling of being transported from the physical location to the virtual location. Immersion and presence are distinct but interrelated concepts. It is believed that systems with a high level of immersion will increase the likelihood of inducing feelings of presence. Furthermore, presence is considered pivotal to the effectiveness of the virtual environment, as it relates to the virtual environment’s ability to fulfill its purpose [26,27,28]. In the case of IVN, this translates into the ability of the IVN to elicit similar responses as interactions with real nature. A recent study provided support for this assumption, showing higher levels of stress reduction and positive affect in an underwater VR experience compared to a desktop counterpart [29].

Unfortunately, highly immersive virtual systems appear to have their limitations. For example, cyber sickness is a well-known side-effect of virtual environments. This malaise is a specific type of visually induced motion sickness [30] and may cause dizziness, nausea and general discomfort. The most prevalent explanation for cyber sickness suggest that the symptoms arise from a sensory conflict between visual, vestibular and proprioceptive signaling [31]. In other words, the visual input from the HMD does not match the input from the surroundings as perceived by vestibular and proprioceptive systems. Cyber sickness is reported to occur in as much as 100% of viewers depending on factors such as the contents of the virtual environment, exposure duration and technological fidelity [32,33,34,35,36,37]. Furthermore, recent research reports that cyber sickness and presence are inversely related, which suggests that cyber sickness may have a negative impact on the feeling of presence in virtual environments [38]. The issue of cyber sickness has recently become more relevant, as visual displays that are considered more immersive, such as HMDs, are more prone to induce high levels of cyber sickness [37]. This paradox must be solved in order to increase the usefulness of IVN, as the most advanced displays may be needed to provide a sufficient degree of presence. Luckily, researchers continue to identify factors that either increase or decrease the levels of cyber sickness, such as habituation, scene oscillation, movement lag and exposure time [34,39,40,41].

Combining IVN and physical activity, a combination that may be defined as virtual green exercise (i.e., physical activity in the presence of technological nature), introduces additional challenges, mainly associated with the issue of maintaining balance and, thus, exacerbating the sensory conflict leading to cyber sickness. In a study of virtual green exercise it was found that cyber sickness had a severe detrimental effect on participants’ emotional responses, which lead to a significant difference in participants’ emotional states after a bout of real green exercise compared to virtual green exercise [39]. Several participants also complained about the difficulties of maintaining balance and reported frustration because their movements did not sufficiently match the movements in the virtual environment [39]. Technology is advancing rapidly, and some of the challenges mentioned above might be addressed in the near future. In fact, a recent study has successfully reduced negative side-effects caused by cyber sickness by minimizing camera oscillations in 360° videos of green exercise [34].

### 2.2. Is Virtual Nature as Good as the Real Thing?

Emerging research on technological nature tentatively confirms that these interactions are more beneficial for health and well-being than an absence of human-nature interactions, but not as beneficial as genuine nature exposure [42,43,44] (see Figure 1). Findings from a range of studies suggest that virtual nature interactions produce some positive effects, but also show that virtual nature is unable to fully reproduce the effects of real nature [44,45,46,47]. Similar reports are found in studies comparing virtual to real green exercise. For example, a recent systematic review reported inconclusive evidence concerning the extent to which virtual green exercise can provide similar psychological or physiological health benefits as real green exercise [48]. The authors of the review warn, however, about limited research rigor in the individual studies, as well as a large variety of outcome measurements and the duration and mode of the physical activity interventions. It should also be noted that among the reviewed studies there was a large variation with respect to the technology used in the virtual nature conditions, with part of the studies using HMDs (only one of which as using a full 360° IVN), while other studies used non-immersive types of virtual nature (i.e., images or videos on a screen). This insight gives hope for the future of virtual nature, as it is expected that increased technological fidelity will improve both immersion and presence, which should improve the psychophysiological effects of virtual nature (see Figure 1). Recent studies adopting modern technology to create immersive nature experiences partly support this notion. For example, Chirico and Gaggiolo [49] created an IVN consisting of a static panoramic video of a natural landscape and successfully replicated some of the positive psychological responses recorded when the participants were exposed to the corresponding real landscape. Similarly, Yu et al. [50] found that exposure to an IVN was effective in eliciting psychological responses similar to those that would be expected in real nature. However the same IVN was unable to produce similar responses with respect to physiological measurements. Browning et al. [51], on the other hand, found a similar physiological response for IVN compared to a real natural environment, but also demonstrated a superior effect on mood levels for the real natural environment.

The aforementioned research suggests that IVN is currently not able to fully reproduce the whole range of psychophysiological responses that people experience in real nature. Further technological and scientific progress may allow the use of highly immersive IVNs to better recreate the fidelity of authentic experiences of nature. IVN may represent a valuable compromise when trying to balance people’s basic need to experience nature with the increasing distance between them and real natural environments, as well as balancing methodological rigor and life-like experiences in a research context. It is likely, however, that, no matter how much the technology improves, IVN will never fully replicate the holistic, multi-sensory and potentially elating experience of the real outdoors. The voice of one of the participants in our VR trials states the following in this regard: “Nature will always win for me. It is less stressful, you know where you are, you can stop and look, for example, at birds anytime.” ([39]; unpublished quote). Moreover, we should bear in mind Levi and Kocher’s warning: “the problem with virtual nature—like the problem with plastic trees—is that the value of nature is more than the experiential and recreational benefits it provides to people. Nature provides a variety of benefits beyond human’s immediate experience; nature exists and has value separate from human beings” [15] (p. 224). In addition, outdoor nature contact offers many additional benefits which to date cannot be incorporated in VR, such as, for example, enhanced immunity from exposure to microbiomes and phytoncides from trees (see [52]). Nevertheless, the opportunity that IVN can provide should not be overlooked, and these possibilities will now be reviewed.

## 3. Enable, Reconnect and Augment

### 3.1. Enable: Virtual Nature as a Supplement to Real Nature

Virtual nature may be used as a nature-based intervention for specific situations where exposure to real nature is difficult or inconvenient. Research has already identified several areas where IVN show particularly promising results and potential applications.

Palliative treatment in clinical settings–IVN has received a lot of attention in the field of clinical care, especially within palliative treatment. The interest in using VR technology as a tool for prevention and treatment of both mental and physiological health issues began in the 1990s [53,54]. Although some researchers have called for high quality studies to demonstrate the cost-effectiveness of VR in clinical settings, research within this field has consistently demonstrated that the use of VR technology is feasible and safe, and results in high patient satisfaction [55]. In a fact, a recent review by White et al. concludes that IVN is a useful tool to integrate with traditional treatment situations in which contact with real nature is not possible or unsafe, e.g., when the risk of injury outweighs the health-promoting effect of real nature [54]. White et al. [54] further reports that IVNs can be effectively applied in the following fields: pain management, neurological disorders, stroke rehabilitation, distraction and relaxation tools in cancer treatment, cognitive rehabilitation and mental health and well-being, including depression, anxiety, obesity, eating disorders and phobias.

Stress-management in the workplace–According to the European Agency for Safety and Health at Work [56], work-related stress is one of the biggest work-related health issues. Interestingly, a recent review of the literature highlights the stress-reducing effects of indoor nature exposure (indoor environments that contain real or representations of nature-based stimuli that engage a variety of senses), and emphasizes that the health benefits of indoor nature exposure occur by facilitating both the reduction of and recovery from stress [57]. IVN technology might further advance the field of nature in indoor settings by generating a more immersive and life-like experience. Although there is a general lack of research regarding the effectiveness of implementing IVNs in the workplace, some research has demonstrated that IVNs can induce stress reduction in experimental trials on healthy adults [50,58,59,60]. Moreover, exposure to IVN has been found to be a more effective tool to reduce anxiety levels and improve mood states when compared to images of nature presented on a traditional computer screen [29].

Mental health and cognitive development in school settings–Today’s children spend a limited amount of time in contact with nature [11,12]. This is unfortunate as research demonstrates that nature contact has a positive impact on both physical and mental health in children and adolescents. For example, Mennis, Mason and Ambrus [61] showed that adolescents with immediate access to greenspaces within their living environment experienced reduced levels of psychological stress. Outdoor recreational activities have also been found to improve adolescents’ self-esteem, well-being and perceived body image [62,63,64]. Nature interactions during childhood can also lead to a greater engagement in nature-based physical activity in adulthood [10,65], and can thus promote lifelong physical activity and improved physical and mental health. Although IVN may not provide all of the beneficial elements of interaction with real nature, it may still provide some benefits, especially with respect to cognitive restoration and enhanced psychological states [66]. There has also been increased interest in using VR as a supplement to children’s school-based education [67]. In this regard IVNs may represent a useful tool to supplement interactions with real nature in schools with limited access to natural areas. VR provides safe environments for pupils and students to learn and gain skills, and IVN might be used to facilitate initial positive experiences of mastery that might reduce children’s fear or insecurities when exploring the outdoors. Inspiration for this particular type of IVN may come from research within the field of nature advertisement and promotion, which provides useful recommendations for designing advertisements to promote green exercise participation in different groups of people [68]. These recommendations can be used to develop tailored IVNs that encourage children to visit real natural environments.

Nature experiences for astronauts in space-missions–In such conditions, IVN represents the only alternative for nature exposure. Space is an extreme environment for the human body and poses a serious threat to human health, as the lack of gravity leads to negative side effects such as muscular atrophy [69]. IVNs might help astronauts coping with the stress associated with this extreme environment as well as augment exercise routines in microgravity. This is an exciting line of research, with recent studies already trying to explore the potential use of psychological interventions to assist in the adaptation to and recovery from exposure to space and space-like environments [70].

### 3.2. Reconnect: Virtual Nature as a Strategy to Reconnect People to Nature

With the backdrop of land use change in the form of rapid urbanization as well as concerns regarding environmental sustainability, the concept of *nature connectedness* (i.e., an individual’s’ feeling of being emotionally connected to the natural world) has emerged as a key aspect of the human-nature relationship. Not only is nature connectedness an important component fostering sustainable behavior [8], but individuals with more positive attitudes towards nature were also found to spend more time in natural environments [10,71,72,73]. This may explain, at least in part, why nature connectedness has been linked with a range of health outcomes. For example, a recent meta-analysis showed that individuals who report higher levels of nature connectedness tend to experience higher levels of positive affect, vitality and life satisfaction compared to those less connected to nature [7]. People who report more positive feelings towards nature were also reported to be more likely to meet the minimum recommended levels of physical activity [10]. Exposure to and interaction with nature seems to be a key element to promote greater feelings of nature connectedness. Childhood experiences of nature, in particular, are known to be a strong predictor of positive feelings towards as well as more frequent interactions with nature as an adult [10,65]. Pupils attending schools with more opportunities for nature contact were found to be more empathetic and concerned for non-human life forms, as well as more aware of human-nature interdependence [74]. Nature experiences can also enhance nature connectedness in adult populations. For example, a tree-planting program was found to enhance participants’ feelings of connectedness with nature, which in turn led to increased engagement in pro-environmental behaviors [75]. Simpler forms of human-nature interactions might lead to similar outcomes. For instance, in a series of experiments by Mayer et al. [76], it was found that a single 10-min walk in a pleasant natural environment can lead to enhanced nature connectedness in college students.

Limited research suggests that exposure to virtual nature may have a similar effect, although scientific research in this field is extremely scarce. Mayer [76] found that participants’ sense of connectedness to nature was improved after watching short videos of pleasant natural environments, with patterns that were similar, although with a smaller effect-size, to an actual walk in real nature. Moreover, connectedness to nature and time spent outdoors are related [10,71,72,73]. Thus, by triggering a greater feeling of nature connectedness, virtual nature may be used as an instrument to re-connect people with nature in a broader sense, persuading them to visit real natural environments, although, to the best of the authors’ knowledge, no study to date has investigated this hypothesis. However, motivational theories are already in place to support the idea. For instance, Calogiuri and Chroni [77] have proposed a model based on the Theory of Planned Behavior [78], which describes that exposure to and experiences in natural environments can influence people’s attitudes towards nature-based physical activity and their future intention and behavior. Positive experiences associated with IVNs might serve as a type of positive reinforcement that can enhance people’s attitudes towards nature-based activities. Research that aims to promote nature-based physical activity has indeed emphasized the importance of presenting natural environments that are considered highly restorative, in line with the Attention-Restoration Theory by Rachel and Stephen Kaplan [68].

### 3.3. Augment: Virtual Nature to Boost the Benefits of Human-Nature Interaction

The effectiveness of virtual nature can go beyond facilitating interactions with simulated nature (Enable) or even helping people reconnect with the real natural world (Reconnect). Virtual nature may offer the possibility to “boost” human-nature interactions (Augment), leading to more restorative experiences as well as enhanced knowledge and engagement. This can be accomplished by the inclusion of virtual elements such as markers to follow pre-set journeys, interactive information-points for learning experiences, guided instructions for meditation, etc. All of these (and many more) components can, rather than simply expose passive viewers to sceneries of nature, trigger and direct users’ attention as well as encourage them to interact with the virtual world and engage in reflections. By using these techniques, virtual nature can maximize restorative experiences. Virtual nature experiences may be designed in a way that users may not only enjoy a pleasant walk in nature, but could also, for example, learn about biodiversity, its importance and how it can be protected (see also [21]).

It should be noted that these possibilities are not a specific prerogative of IVN, as they also apply to other forms of non-immersive virtual nature. Non-immersive virtual reality or ‘mixed reality’ might be used for the same purposes and have the advantage of being easily integrated into people’s everyday routines. This includes, for example, mobile apps and augmented reality (AR). AR technology, in particular, has been emerging as a valuable supplement to traditional education tools [79] as well as to engaging visitors to touristic locations, including natural parks [80]. More specifically, AR applications have been shown to engage its users in natural environments as well as teach them about environmental issues such as water quality and biodiversity to a greater extent than other educational tools do [81,82].

On the other hand, given IVN’s potential to provide more immersive experiences and more life-like perceptions, as well as its greater effectiveness in eliciting psycho-cognitive restoration as compared to non-immersive virtual nature [29] and other virtual experiences [66], IVN may have the advantage of engaging users’ attention to a greater extent and may lead to more intense emotional responses. By disconnecting the viewer from the external (real) world and creating a limited and controllable (virtual) environment, IVN provides particularly favorable conditions in which nature-savoring (i.e., a person’s ability to attend to, appreciate and enhance the positive experience of being in contact with nature; [83]) can be triggered and trained. Special IVN-based programs can be designed to elicit and train users’ nature-savoring, an ability they can later apply in the presence of real nature, maximizing the psychological benefits of the human-nature interaction. Thus, nature-based interventions can not only provide an opportunity for the experience of nature but can in addition provide specific learning on psycho-social skills or sustainability which can optimize future human nature interactions [21].

Regarding how virtual nature can boost users’ nature experiences, gamification is another important concept. Gamification is defined as “the use of game design elements in non-game contexts” [84] (p. 2) and has the purpose to motivate and increase user engagement. While gamification has been predominantly applied in business, marketing and corporate management, its use is also increasing in the field of education [85] and health promotion [86]. Gamification might be included within virtual nature systems to, for example, enhance users’ compliance with psychological training programs (e.g., by adding rewards or competition elements) as well as enhance their engagement in educational processes (e.g., through quizzes).

## 4. Future Perspectives for Research on (or Involving) Virtual Nature

This section will discuss the potential of IVN as a nature-based intervention in addition to evaluating the methodological challenges and providing specific recommendations for researchers in this field. Researchers within areas such as environmental psychology and public health have an interest in understanding how natural environments affect human emotions, cognition, behaviors and health. Such an understanding may have several uses, including designing outdoor or indoor environments that induce stress recovery and helping policy makers and planners take informed decisions about regulations related to planning and re-naturing public spaces [87]. This research can also help in understanding the psychophysiological mechanisms underlying human-nature interactions, as well as engagement in pro-environmental behaviors and sustainable lifestyles. However, examining how people respond to natural environments, as compared with other indoor or outdoor environments, is a difficult task. Conducting rigorous research requires that possible confounders are eliminated or controlled for; this can be accomplished by conducting experimental trials in standardized environmental conditions (laboratory-based studies) and by performing appropriate randomization and blinding procedures, which is often challenging or even impossible to accomplish in natural settings. Thus, in this context, VR technology might be particularly useful. Despite being a young research area, there are several known factors that should be considered when conducting studies using IVN, and more can be deduced from theoretical frameworks. Based on these factors, the following sections will present a generic methodology to consider when planning and conducting research with IVNs based on 360° 2D images/videos to be presented using HMDs.

### 4.1. General Considerations

#### 4.1.1. Study Design

A recent systematic review of the literature [48] found that studies of green exercise, including virtual green exercise, are often characterized by a high risk of bias due to (I) inadequate/unclear randomization procedures; (II) a lack of blinding of both participants and assessors to the experimental conditions; (III) inadequate washout periods in trials with crossover design; (IV) and potential contamination in control conditions. Furthermore, most trials had insufficient statistical power, and the scarcity of preregistered trials limited the possibility of ruling out selective reporting. Thus, it is recommended that future research on (or involving) IVN be conducted according to general guidelines for randomized controlled trials [88]. Accordingly, performing appropriate randomization procedures, as well as the blinding of participants and possibly also the examiner, is paramount to reduce risks of bias in the assessments and analyses.

When planning trials with a crossover design, it is important to consider that the carry-over effect is a real concern in studies involving IVNs. This may relate especially to negative emotional and physical responses associated with cyber sickness. A review by Duzmanska [40] indicated that symptoms of cyber sickness may last up to 4 h depending on the severity of the symptoms and the duration of exposure. Thus, it is recommended that trials with a crossover design avoid administering multiple IVN exposures within the same day.

The characteristics of the control and/or comparison conditions also require careful evaluation. Some studies have used “true control conditions” such as sitting quietly staring at a blank wall (e.g., [51]). This may, however, be problematic. Whitehead [89] recommends using an “active control” condition, in general, to sufficiently control for a potential placebo effect. This may be particularly important when the IVN intervention involves physical activity, which can induce psychological benefits in itself [90]. Furthermore, the true control condition may elicit feelings of boredom, which may in turn lead to negative emotional states. For example, in a summary of 11 studies, Wilson et al. [91] reported that sitting alone in a room doing nothing was perceived as non-enjoyable, to the point that in one of the reported studies many preferred to administer mild electric shocks to themselves. Many studies involving IVN or other forms of virtual nature have rather opted for comparison conditions that retained some similarity with the treatment condition, e.g., exposure to virtual urban environments [2,48,92].

#### 4.1.2. Theory-Based Approaches

Theory-led investigations of virtual nature may be useful for advancing our understanding of the mechanisms underlying benefits to health from nature contact. A multiplicity of possible mechanisms have been proposed, but the majority of these assume direct rather than indirect (e.g., virtual) nature contact [4,21,52]. A recent review has attempted to explain the benefits and risks of human-nature interactions by accounting for dose-response relations, exposure (referring to the amount of contact that an individual or population has with nature), experience (includes interaction and dose), and natural features [4]. More generally, contemporary conceptual frameworks [4,42] can help generate additional testable hypotheses beyond the traditional approaches of the Stress-Reduction Theory [93] and Attention-Restoration Theory [94].

#### 4.1.3. Natural Typologies

The presence, distribution and diversity of natural features in the environment, whether of a virtual or authentic nature, have to be considered [95,96]. Features of nature that potentially influence mental health include the size of the environment (total area), its composition (proportions of different types of natural elements) and spatial configuration (e.g., degrees of fragmentation and connectivity with other green spaces) [95,96]. Other relevant natural attributes may include the tree canopy density, vegetation structure, species composition or biodiversity across a range of different settings or typologies. These typologies include public green spaces, peri-urban nature reserves, wilderness and pastoral landscapes [97]. Preferences for different typologies should be considered in developing IVN stimuli. As Depledge et al. [98] suggested, “we are ignorant of how subtle changes in these features, or the removal of certain sensory effects from an environment (“de-integration”), are perceived by simulation users and what effects these may have on performance, engagement and, indeed, well-being” (p. 4463). VR can provide a laboratory to dissect at least some of the stimuli we receive from being outdoors in nature and to aid the evaluation of their relative contribution to well-being.

#### 4.1.4. Reporting Findings

The systematic review by Lahart et al. [48] calls for enhanced standards in reporting green exercise (and virtual green exercise) studies. In line with such a call, the authors of this review encourage researchers who perform IVN studies to report their findings in compliance with internationally recognized guidelines (e.g., CONSORT statement). In particular, Lahart et al. [48] encourage researchers to provide clear information regarding randomization procedures and methodological transparency and rigor via the preregistration of study designs and statistical plans, as well as by making data openly accessible.

In addition to these general recommendations, it is recommended that researchers report details of the IVN technology used in their studies (brand and model of devices, hardware and processing techniques), as well as detailed information on how the IVNs were developed, as research suggests that the generalization between devices may be limited [99,100]. Information about the participants’ characteristics, especially in relation to possible confounding variables (see Section 4.2.3), should also be clearly stated when reporting findings.

### 4.2. Specific Considerations

#### 4.2.1. Duration of the IVN Exposure

The exposure duration should be carefully evaluated, as this might influence both the effect size of the psychophysiological responses to virtual nature and influence the risk of cyber sickness. Shorter bouts (e.g., 5 min) of nature experiences have been associated with the largest effect sizes on self-esteem and total mood, while benefits on biological indicators of stress (e.g., blood pressure) would peak at 10 min of exposure [62]. On the other hand, in IVN studies it is important to take into consideration the possible impact of the IVN exposure on cyber sickness. In this regard, a review by Duzmanska et al. [40] shows that cyber sickness symptoms generally increase with time, at least until a certain threshold (~75 min). This means, for specific studies, that there may be a trade-off between the optimal exposure duration to induce the desired effect and the optimal exposure duration to minimize the impact of cyber sickness. In these cases, it may be crucial to conduct a pilot study in advance to identify the ideal compromise.

#### 4.2.2. Choosing the Appropriate Type of IVN

Different types of IVN exist, which differ in the way they are created as well as in their resemblance with real nature. One way to create IVN is to use photorealistic representations using video-game development technology. This type of IVN can achieve relatively high levels of realism, and has been found to elicit psychological and cognitive restoration [60,101]. To the best of our knowledge, however, this technology has not yet been applied to dynamic 360° systems. Another way to create IVN is to use special 360° cameras. This alternative has become increasingly popular as commercial 360° cameras are becoming more affordable and of a higher quality (higher resolution, better in-built stabilization option, greater possibility to edit images and videos in post-production, etc.), making it relatively easy to create IVNs based on existing locations. This technique allows the creation of different types of IVN, each coming with advantages and disadvantages. 360° images may be considered as not being very life-like, as they does not display any moving objects (e.g., leaves moving in the wind or waves rolling). On the other hand, the lack of movement makes 360° images less likely to induce sensory conflict, and thus less likely to induce cyber sickness [31]. Some studies have applied this technique (see e.g., [50]) and show that 360° nature images were able to elicit some of the expected benefits of nature exposure. Static 360° videos are similar to the 360° image, but they also displays the movement of objects, which may make the IVN more ‘life-like’. This technique has also been applied in previous studies (see e.g., [49]), finding that static 360° videos can induce psychological responses similar to those experienced when exposed to real nature. Dynamic 360° videos (i.e., 360° videos in which the viewer perspective moves in the virtual space) may provide the advantage of recreating the feeling of exploring a natural environment and can be administered in combination with physical activity, which is shown to improve the users’ feelings of presence [38], but these videos also have some issues. In particular, they are more prone to inducing cyber sickness, which can have a negative impact on the restorative benefits of the IVN exposure [39]. In this regard, the stability of the sceneries in dynamic 360° videos seems to be crucial for avoiding cyber sickness [34,102,103]. Improved stability can be achieved by using external or in-built stabilizers in combination with a dolly or a hoverboard [34], and available computer software can further reduce vibrations and oscillations in 360° videos. Research also suggests that the posture of the viewer should match the perspective in the virtual environment, i.e., seated participants experience lower levels of presence compared to standing participants when exposed to an environment where you would typically be walking [33]. To further reduce sensory conflict and optimize the potential of dynamic 360° videos, it is important that the pace of the user matches that of their virtual avatar [39]. This can be achieved either by externally controlling the users’ pace (e.g., making them walk on a treadmill at a predetermined speed that corresponds to the speed in the video) or connecting the locomotion device (e.g., a manually driven treadmill or a cycle-ergometer) to the IVN system so that the user can determine the pace. Both options come with advantages and disadvantages, e.g., externally controlling the users’ pace may allow for a better standardization in experimental trials, but allowing the users to determine their own pace may increase the sense of presence.

#### 4.2.3. Control of Confounders

In general, it is recommended that studies involving IVN be conducted in standard environmental conditions, in order to control for general environmental confounders such as temperature, humidity, lighting, etc. More specifically, the literature on VR and IVN has provided evidence on a variety of user-related as well as technology-related factors that influence the way people experience and respond to IVN exposure. In studies involving IVN, it is thus recommended that the researchers control for as many of these possible confounders as possible (e.g., by considering them when outlining the eligibility criteria or by including them as covariates in the statistical analysis), or at least take into consideration the possible impact that they may have on the study outcomes. Some of the key factors are reviewed below.

##### Individual Characteristics

To assess the effects of virtual nature it is important to evaluate how these effects are moderated according to individual differences. Sex has been found to influence both cyber sickness and presence in virtual environments, with men tending to experience lower levels of cyber sickness and higher levels of presence [38]. Studies have identified other factors that may impact cyber sickness, such as genetics [104], habituation [40], visual acuity [32] and postural control [105].

An individual’s habitual physical activity levels have been recently found to influence the extent to which a person is able to correctly estimate the visual speed of their avatar in a non-immersive virtual environment [106]. In particular, as compared with people with more running experience, people with lower levels of weekly physical activity were found to underestimate the visual speed relative to their actual running speed more often. Noticeably, the underestimation of visual speed was dependent on the actual speed (i.e., it was larger at higher speeds). This could be explained by the fact that individuals who regularly engage in physical training show better functions related to visual skills [107,108,109]. Although this phenomenon has been, to the best of our knowledge, shown only in non-immersive virtual environments, it is likely that similar patterns could also be observed in immersive virtual environments, such as dynamic IVNs. In this regard, some participants may experience a feeling of mismatch between the actual walking/running/cycling speed on an ergometer and the speed in the IVN, even though the speeds are designed to match, potentially leading to a reduced sense of presence and increased levels of cyber sickness.

Personality has also been suggested as a factor that can influence how individuals experience and respond to virtual environments. While it is established that individuals’ personality influences the way they perceive and respond to different environments, recent studies have also found similar patterns when individuals are exposed to IVNs (e.g., [110]). It has been proposed that presence might play an important role in explaining the relationship between personality and the inter-individual differences in VR experiences. However, this relationship was found to be largely dependent on the instruments used to assess both personality and presence [111]. One individual factor that should be considered is nature connectedness, which can easily be examined with standardised instruments (e.g., NR6; [112]). As outlined above, nature connectedness influences people’s inclinations towards nature, and this may extend to virtual nature. Similarly, it may be useful to assess other psychological factors which have been subject to recent study in the broader green exercise literature. For example, Flowers et al. [113] validated a tool to assess beliefs about green exercise which can help account for placebo effects.

Novelty and habituation may also influence the results in studies of IVN. Habituation reduces the impact of cyber sickness [40], which suggests that participants who are familiar with VR may perceive IVN as a more positive experience. On the other hand, recent research reports that the novelty effect also influences the results in studies on virtual and augmented reality [114,115], which suggests that participants who are unfamiliar with the technology might view the experience as positive based on the novelty of the technology alone. Although a study of IVN by Browning et al. [51] concluded that the novelty effect did not influence the results in studies of virtual nature, it may be too soon to make a definite conclusion regarding this matter. Demographic factors including age and being a ‘digital native’ should also be considered depending on the goal of the study and the target sample.

##### Characteristics of the IVN

With respect to recommendations regarding appropriate technology in IVN trials, Rebenitsch and Owen [99] provide an exhaustive list of display characteristics that may influence the users’ experiences, and particularly cyber sickness. In general, high-resolution IVN systems would be preferable, as qualitative reports indicate that a low resolution or sharpness of the image can be associated with discomforts during the exposure [39]. The frame rate and latency are also shown to influence both cyber sickness and presence [38]. In addition to display characteristics, the quality of the sound also needs to be cautiously considered. Previous research shows that the sounds and acoustics in both real and virtual environments have a prominent impact on people’s experiences. The results from a pilot study by Annerstedt et al. [116] showed that virtual nature with sounds was effective in eliciting stress-reducing effects, while the effect was absent in virtual nature without sounds. Research on soundscapes has demonstrated that auditory input can influence both psychological and physiological measures of health and well-being [117,118,119]. Soundscape refers to acoustic environments as perceived by people, in context, and evidence is accumulating to support the proposition that urban soundscapes contribute to the environmental quality of urban areas in the same way that microclimatic data does [120]. There are different techniques for creating sounds in IVN, and some have been proven to create more realistic experiences as compared to others (for an overview on this issue, see [121]). This accumulating evidence suggests that the type and quality of the soundscape is important to consider when creating IVNs.

## 5. Limitations

The main limitation of this review is the narrative nature of the paper, as well as the fact that some aspects that relate to IVN are only discussed briefly. However, a full systematic review and detailed description of all aspects related to IVN is a monumental task and beyond the scope of this paper. The focus of this review was to give an overview of key areas related to the creation and implementation of IVNs. In addition, because VR, IVN and related research areas are still in their infancy, it is expected that there are still several unknowns in the process of successfully implementing IVNs both in research and in real-life situations.

## 6. Summary and Conclusions

Nature experiences in their various forms will continue to provide a pathway to enhanced well-being and health, and the contribution of technological nature, and specifically IVN, has yet to be fully elucidated. This paper suggested several uses for IVN. Some uses are backed by substantial research, while other uses still await solid confirmation or disconfirmation. Nevertheless, the diminishing access to authentic nature in urban settings should be a driver for further exploration of IVNs’ ability to enable, reconnect and augment human-nature experiences. IVN as a research area, and the technology it involves, are still in its infancy, which suggests that there is an untapped potential that might be uncovered in the future. To tap into this potential, researchers and manufacturers must identify strategies to deliver highly immersive experiences with high levels of presence, while at the same time avoiding the issue of cyber sickness. In this regard, specific recommendations for the next wave of research have been provided. Caution is advised, though, as concerns have been raised regarding the risk of replacing real nature with virtual nature, and thus accelerating the disconnection from the natural world. The long-term effects of using IVN are also unknown; it is possible that the positive effects identified by short-term studies will diminish in the long term.

## Figures and Tables

**Figure 1 ijerph-17-01738-f001:**
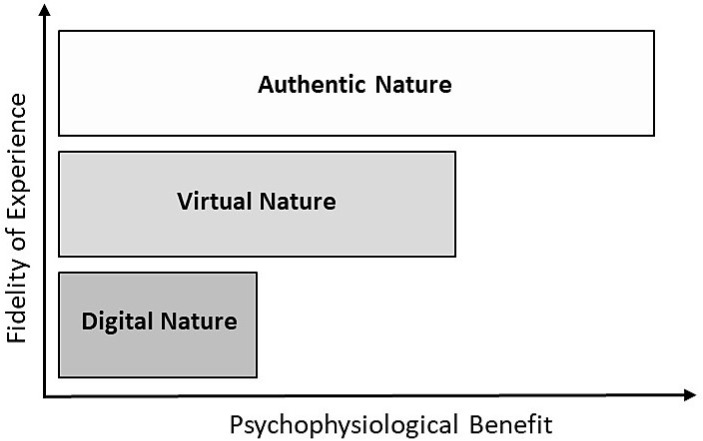
Predicted psychophysiological benefits from nature experiences at different levels of fidelity.

**Table 1 ijerph-17-01738-t001:** Definitions of the key terms and key concepts relating to immersive virtual nature.

Term	Definition	Source
Nature	Refers to “areas containing elements of living systems that include plants and non-human animals across a range of scales and degrees of human management—from a small urban park to “pristine wilderness.”	[19] (pp. 121–122)
Green exercise	“Physical activities whilst at the same time being directly exposed to nature.”	[20] (p. 7)
Nature-Based Interventions	“NBIs are programmes, activities or strategies that aim to engage people in nature–based experiences with the specific goal of achieving improved health and wellbeing.”	[21] (p. 2)
Virtual Reality (VR)	“A medium composed of interactive computer simulation that senses the participant’s position and actions and replaces or augments the feedback to one or more senses, giving the feeling of being mentally immersed or present in the simulation (a virtual world).”	[22] (p. 13)
Augmented Reality (AR)	AR “supplements the real world with virtual (computer-generated) objects that appear to coexist in the same space as the real world.”	[23] (p. 34)
Immersion	“The extent to which the computer displays are capable of delivering an inclusive, extensive, surrounding and vivid illusion of reality to the senses of a human participant”	[24] (p. 3)
Presence	‘‘The (psychological) sense of being in the virtual environment.”	[24] (p. 3)
Technological Nature	“Technologies that in various ways mediate, augment or simulate the natural world.”	[14] (p. 37)
Immersive Virtual Nature (IVN)	Based on so-called immersive virtual environments technology, provides the illusory perception of being enclosed within a natural environment.	[17] (p. 280)

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
