# Peer review of "Enable, Reconnect and Augment: A New ERA of Virtual Nature Research and Application"

_ijerph, 2020, doi:10.3390/ijerph17051738_

Round 1

Reviewer 1 Report

Thank you for the opportunity to read and review the manuscript “Enable, Reconnect and Augment: A new ERA of  virtual nature research and application”

In this manuscript the authors analyze the technological nature influence on human nature and public health. Authors discuss the role of IVT in enabling, reconnecting and augmenting human interactions with real nature. Overall the paper is interesting and suitable for International Journal of Environmental Research and Public Health.

The paper is interesting and ready to be published.

The analyzed issue is presented in clear and coherent manner.

The literature references are up-to-date and relevant.

My only concern refers to lack of limitation. Authors should indicate possible shortcomings of this paper.

Additionally, in my opinion, some contribution should be provided.

Author Response

Thank you very much for your positive feedback and valuable input on how to improve the paper.

Regarding point 1: My only concern refers to lack of limitation. Authors should indicate possible shortcomings of this paper

Response 1: We agree to your concern, and we have now added a limitations section in lines 572-579

Regarding point 2: Additionally, in my opinion, some contribution should be provided.

Response 2: We have not included a contribution section in this paper, but we have identified contributions in the Summary and conclusion section. I.e. line 584 “This paper have suggested several uses for IVN”, Line 591 “…specific recommendations for the next wave of research has been provided”. We believe that this is appropriate, but please comment further if you would like us to include clearly defined contributions. 

Reviewer 2 Report

The authors investigated an interesting subject considering the importance of nature on human mental health and increasing screen time. Overall the paper is well-written and convincingly outlined the idea of virtual nature. As Ulrich and other researchers point out, perceived restorativeness can be provided by nature and green spaces. Integrating restorativeness in to screen will help those who have little access to nature and green environment. In addition, there are not much studies to investigate virtual nature. Hence, this paper is worth for publication. I only have one minor suggestion is that the purpose of the paper should be clearly stated in the paper.     

Author Response

Thank you very much for ypur positive feedback and valuable input on how to improve this paper.

Regarding point 1:  I only have one minor suggestion is that the purpose of the paper should be clearly stated in the paper.

Reponse 1: We have added a clear statement of the purpose of this paper in lines 85-86 

Reviewer 3 Report

This paper addresses an emerging topic in the research area of nature and human health. It is well-written and provides sufficient information on the pros and cons of this technology. The recommendations suggested by the authors provide a guideline/standard to move this field of research forward. I consider this paper to be deserved for publication. I only have a couple of minor comments and most of them are editorial. The comments I have are detailed below.

1) virtual nature cannot provide many types of ecosystem services (e.g., heat mitigation, biodiversity), which may, in turn, affect human psychological or physiological health. This limitation should also be cautioned.
2) "2.3 Augment: virtual nature to boost the benefits of human-nature interaction" should be "3.3"
3) Line #382, please add the citation for the sentence "A recent review..."
4) Citations are needed for section 4.1.3. Natural Typologies

Author Response

Thank you very much for your positive feedback and valuable input on how to improve the paper.

Regarding point 1: virtual nature cannot provide many types of ecosystem services (e.g., heat mitigation, biodiversity), which may, in turn, affect human psychological or physiological health. This limitation should also be cautioned.

Response 1: We have added a warning regarding the inability of virtual nature to provide important ecosystem services in lines 73-75

Regarding point 2: "2.3 Augment: virtual nature to boost the benefits of human-nature interaction" should be "3.3"

Response 2: This has been changed to 3.3. See line 300.

Regarding point 3: Line #382, please add the citation for the sentence "A recent review..."

Response 3: Citation added. See line 401.

Regarding point 4: Citations are needed for section 4.1.3. Natural Typologies

Response 4: We have added two citations in line 406 and 410.

Reviewer 4 Report

The presented review is exploring a very interesting argument. Beside the conclusions and discussion reported by the authors, I believe there will be some very useful application of green virtual reality to specific group of population such as elderly or people who cannot move from home, but in the meantime, they can benefit from a positive virtual reality. Thus, studying this subject would be very useful for innovation and future. Some suggestion are reported to the authors to improve the document quality.

I suggest avoiding the use of “we” in scientific writing. At least, please use a uniform style trough the document: in the introduction “we” is used, while (e.g.) at line 152 it is used “The authors of the review…”, as I sincerely prefer.

Figure’s 1 quality is low. Please increase it.

Chapter 3 has title which is almost the same as the paper itself. This sounds bad and I suggest to change it.

Line 533. I was expecting earlier to find something about noise and acoustical environments. I believe that topic deserves more space. At first, at list in the introduction I would use at least one line for mention that both visual and acoustic aspects are closely related and both are important to be considered. Furthermore, around line 533 it is important to dedicate more space to the acoustics and soundscape area with more reference. Soundscapes effects have been recently studied by many authors and the importance good environments as quite areas have been demonstrated (Cassina, L., Fredianelli, L., Menichini, I., Chiari, C., & Licitra, G. (2018). Audio-visual preferences and tranquillity ratings in urban areas. Environments, 5(1), 1.; Aletta, F., Oberman, T., & Kang, J. (2018). Associations between positive health-related effects and soundscapes perceptual constructs: A systematic review. International journal of environmental research and public health, 15(11), 2392; Erfanian, M., Mitchell, A. J., Kang, J., & Aletta, F. (2019). The Psychophysiological Implications of Soundscape: A Systematic Review of Empirical Literature and a Research Agenda. International journal of environmental research and public health, 16(19), 3533.). In fact, I believe that reproducing in the VR speakers a proper and relaxing sound, coherent with what is played on video, would enhance the virtual experiences.

Author Response

Thank you very much for your positive feedback and valuable input on how to improve the paper.

Regarding point 1: I suggest avoiding the use of “we” in scientific writing. At least, please use a uniform style trough the document: in the introduction “we” is used, while (e.g.) at line 152 it is used “The authors of the review…”, as I sincerely prefer.

Response point 1: We agree. We have some minor changes throughout the manuscript to avoid the term “we” when referring to the authors of the paper. See lines 21, 23, 76, 141, 361, 422, 428.

Regarding point 2: Figure’s 1 quality is low. Please increase it.

Response point 2: We agree. The quality of figure 1 has been increased. See line 181.

Regardig point 3: Chapter 3 has title which is almost the same as the paper itself. This sounds bad and I suggest to change it.

Response 3: We agree. The title has been changed to "Enable, Reconnect, and Augment". See line 202.

Regarding point 4: Line 533. I was expecting earlier to find something about noise and acoustical environments. I believe that topic deserves more space...

Response 4: We agree. We have added one line in the introduction to clarify that IVN consists of both auditory and visual stimuli. See lines 66-67. We have also ephasized the importance of soundscapes in lines 558-559, 562-563 and lines 568-569. We have also added the suggested citations. However, as outlined in the newly added limitations section we do not intend to give an in depth explanation for all aspects related to immersive virtual nature. The visual aspect has received additional attention in this review because of the recent wave of head-mounted displays and the relation between visual stimuli, cyber sickness and ability to maintain balance during upright exposure to immersive virtual nature.